# Whole-genome sequencing analysis of the cardiometabolic proteome

Arthur Gilly[1,2], Young-Chan Park[2,3], Grace Png[1,2], Andrei Barysenka[1], Iris Fischer [1], Thea Bjørnland[2,4], Lorraine Southam[1,2,5], Daniel Suveges[2,6], Sonja Neumeyer[1], N. William Rayner[1,2,7,8], Emmanouil Tsafantakis[9], Maria Karaleftheri[10], George Dedoussis[11] & Eleftheria Zeggini [1,2,12 ✉]

The human proteome is a crucial intermediate between complex diseases and their genetic and environmental components, and an important source of drug development targets and biomarkers. Here, we comprehensively assess the genetic architecture of 257 circulating protein biomarkers of cardiometabolic relevance through high-depth (22.5×) whole-genome sequencing (WGS) in 1328 individuals. We discover 131 independent sequence variant associations ($P < 7.45 \times 10^{-11}$) across the allele frequency spectrum, all of which replicate in an independent cohort ($n = 1605$, 18.4$x$ WGS). We identify for the first time replicating evidence for rare-variant $cis$-acting protein quantitative trait loci for five genes, involving both coding and noncoding variation. We construct and validate polygenic scores that explain up to 45% of protein level variation. We find causal links between protein levels and disease risk, identifying high-value biomarkers and drug development targets.

[1] Institute of Translational Genomics, Helmholtz Zentrum München—German Research Center for Environmental Health, Neuherberg, Germany. [2] Wellcome Sanger Institute, Wellcome Genome Campus, Hinxton CB10 1SA, UK. [3] University of Cambridge, Cambridge, UK. [4] Department of Mathematical Sciences, Norwegian University of Science and Technology, NO-7491 Trondheim, Norway. [5] Wellcome Centre for Human Genetics, Oxford, UK. [6] European Bioinformatics Institute, Wellcome Genome Campus, Hinxton CB10 1SH, UK. [7] Wellcome Centre for Human Genetics, Nuffield Department of Medicine, University of Oxford, Oxford, UK. [8] Oxford Centre for Diabetes, Endocrinology and Metabolism, Radcliffe Department of Medicine, University of Oxford, Oxford, UK. [9] Anogia Medical Centre, Anogia, Greece. [10] Echinos Medical Centre, Echinos, Greece. [11] Department of Nutrition and Dietetics, School of Health Science and Education, Harokopio University of Athens, Moschato, Greece. [12] TUM School of Medicine, Technical University of Munich and Klinikum Rechts der Isar, Munich, Germany. ✉email: eleftheria.zeggini@helmholtz-muenchen.de

Cardiometabolic diseases are a leading cause of death and continue to rise in prevalence across global populations. Genome-wide association studies (GWAS) have identified large numbers of susceptibility loci. However, the precise biological networks through which these genetic instruments exert their effects remain largely unknown. The genetics of intermediate traits, such as proteomics, is poised to offer insights into disease-relevant mechanisms and pathways, point to new drug targets and identify new biomarkers to improve early detection and diagnosis. GWAS of protein traits have recently become viable[1–6], but imputed GWAS chip data only offers partial insights into the genetic architecture of protein traits, in particular with respect to capturing rare variation. Large studies with proteomic data coupled to high-depth whole-genome sequencing, which is required to study the role of rare variation[7,8], are currently lacking. Here, we perform whole-genome-sequence-based association analysis between 257 cardiometabolic disease-related serum protein levels[9] and 13,419,876 single nucleotide variants (SNVs) in a population-based cohort (MANOLIS), and assess colocalization, causation and predictive power of protein quantitative trait loci (pQTL) in cardiometabolic disease. We report new single-variant associations at 50 protein quantitative trait loci, including from variants with a significantly increased frequency in this isolated cohort. We find robust, replicating evidence of burdens of rare variants, both regulatory and coding, influencing protein levels at five cis loci (ACP6, PON3, IL1RL1, DPP7, CTSO) independently of common variant signals. We describe three loci where causal evidence for protein-disease association is supported by functional information. Finally, we demonstrate that predictive models of hypercholesterolemia are significantly improved by the inclusion of polygenic information from multiple proteins, highlighting the contribution of genetically determined protein levels to cardiovascular disease risk.

## Results and discussion

**Genetic architecture of protein quantitative trait loci.** We identify 116 protein quantitative trait loci (pQTLs) reaching study-wide significance ($P < 7.45 \times 10^{-11}$) (Supplementary Data 1, Fig. 1). Thirty-two (27%) of these are driven by multiple independent variants (between two and seven per locus (Supplementary Fig. 1), giving rise to a total of 164 independently-associated variants, illustrating complex allelic architecture at pQTLs. We find replicating evidence for association ($P < 0.000305$) across 131 out of 159 variants (82%) present in an independent, whole-genome-sequenced population-based cohort with the same serum biomarker measurements (Pomak[10]) ($n = 1,605$, 18.4x WGS). Replication was expectedly poorer for rare variants (Supplementary Fig. 2). We find that these robustly-replicating loci explain up to 47.7% of protein level variance, and on average more (one-sided Mann-Whitney-U test, $P = 3.42 \times 10^{-13}$) than for 37 other, non-proteomic quantitative traits measured in the same individuals (Supplementary Fig. 3). This exemplifies how the study of blood biomarkers can powerfully capture the heritable component of biological processes underpinning such disease-relevant quantitative traits. Eighty of the associated variants display significant allele frequency differences between the isolated population studied here and large reference populations (Supplementary Data 2), 53 of which have increased frequencies in MANOLIS (66%, $P = 0.002$, one-sided 1-sample proportion test). In particular, 14 associated variants display a frequency increase of more than 5-fold, highlighting the advantage of using isolated populations in associations of protein levels.

Ninety percent of these reproducibly-associated variants are common (minor allele frequency (MAF) > 5%), and 76% are located within 1 Mb of the gene encoding the respective protein (i.e. in cis-pQTLs) (Fig. 2). Among these cis loci, thirty-two out of 72 cis-pQTLs (44%) discovered in this cohort have either not previously been reported in protein-level GWAS (novel loci), or harbour variants conditionally independent of all previously-reported associations (novel variants at known loci) (Supplementary Data 1).

We identify 38 variants in 35 trans loci associated with 32 proteins; 18 of these variants both have not been previously reported (Supplementary Note 1) in protein-level GWAS and replicate in the Pomak population. We find the overall replication rate to be similar for trans- (81%) and cis-associated (79%) variants. Of the replicating 31 trans-acting variants, 30 are common and one is low-frequency. We identify trans-pQTL signals for seven receptor/ligand pairs with experimental evidence of physical interaction and well-established synergistic roles in downstream pathways[11–17].

**Rare regulatory variants affecting protein levels.** To enhance our understanding of rare variant (RV) contribution to serum protein biomarker levels, we performed gene-based burden analysis across coding and noncoding sequence variation (Methods). We identify for the first time 6 study-wide significant ($P < 7.45 \times 10^{-11}$) cis-RV-pQTLs (Fig. 3, Supplementary Fig. 4), in the ACP6 (lysophosphatidic acid phosphatase type 6, $P_{\text{meta-analysis}} = 3.17 \times 10^{-97}$), PON3 (paraoxonase 3, $P_{\text{meta-analysis}} = 7.42 \times 10^{-86}$), IL1RL1 (interleukin 1 receptor like 1, $P_{\text{meta-analysis}} = 2.15 \times 10^{-58}$), DPP7 (dipeptidyl peptidase 7, $P_{\text{meta-analysis}} = 2.71 \times 10^{-36}$), CTSO (cathepsin O, $P_{\text{meta-analysis}} = 2.27 \times 10^{-33}$) and GRN (progranulin, $P_{\text{MANOLIS}} = 3.16 \times 10^{-12}$) genes. All except the GRN burden signal replicate in the Pomak cohort. The GRN RV-pQTL is driven by the novel splice donor variant chr17:44349552 (G>A, minor allele count MAC = 4) and the 5′-UTR variant rs563336550 (MAF = 1.7%) in MANOLIS, the latter showing a 17-fold increase in frequency in MANOLIS compared to gnomAD non-Finnish Europeans (MAF = 0.1%), and more than 2000-fold compared to TOPMed (MAF = 0.00079%).

We find that rare regulatory variants are major contributors to some of these burdens. For example, one of the two variants driving the PON3 cis-RV-pQTL resides in promoter ENSR00000215353 and transcription factor binding site ENSR00000832511, and is associated with a decrease in PON3 levels (rs149867961, MAF = 3.1%, effect size $\beta = -1.18$ in units of standard deviation, standard error $\sigma = 0.113$, $P = 7.58 \times 10^{-23}$). The other contributing variant, rs772677677 (MAF = 1.9%, $\beta = -1.55$, $\sigma = 0.143$, $P = 3.15 \times 10^{-24}$), is a missense variant with a substantially increased frequency in MANOLIS (MAF = 1.9% compared to 0.00264% in gnomAD), also associated with a decrease in PON3 levels. PON3 (paraoxonase 3) inhibits the oxidation of low-density lipoprotein (LDL), an effect that slows atherosclerosis progression[18]. These findings illustrate the contribution of rare variants to the heritability of proteomic traits, and that this contribution is partly mediated through cis-RV-pQTLs.

**Identifying causal associations between proteins and disease.** To detect proteins that may play a causal role in cardiometabolic disease onset or progression, we performed two-sample Mendelian randomization analysis across 93 proteins with study-wide significant signals here and 193 diseases and traits from UK Biobank and other large consortium datasets (Methods, Supplementary Data 3a, b). We identify significant (FDR < 0.05) associations involving 48 proteins and 75 phenotypes (Supplementary Data 4, Fig. 4). For 13 of these proteins, pQTL SNPs had both a lowering effect on circulating levels and a protective effect against at least one disease (Supplementary Data 5), suggesting potential antibody-based approaches for therapeutic benefit.

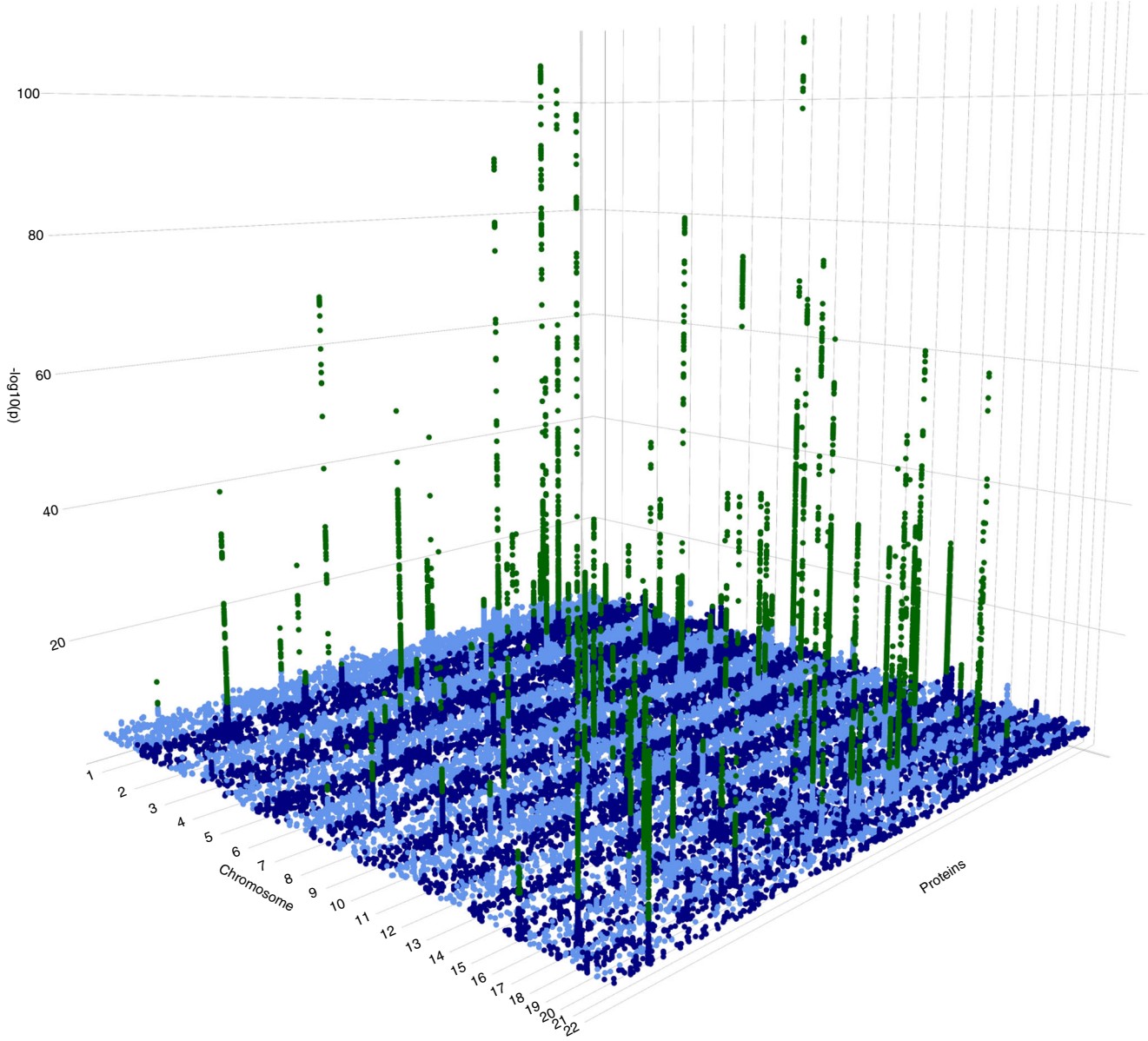

**Fig. 1 Genome-wide association signals across all tested proteins.** For clarity, variants with $P > 1 \times 10^{-5}$ are not represented in the figure. Variants with $P < 7.45 \times 10^{-11}$ are plotted in green. Source data are provided as a Source Data File (score test, one-sided).

Providing proof of principle, we find evidence of established links between dysregulated protein levels and common diseases, such as an inverse causal correlation between PCSK9 levels and hypercholesterolemia ($P = 1.00 \times 10^{-10}$, $P_{FDR} = 9.46 \times 10^{-8}$), and between osteopontin levels and risk of osteoporosis ($P = 6.49 \times 10^{-5}$, $P_{FDR} = 0.011$) and hypothyroidism ($P = 1.2 \times 10^{-4}$, $P_{FDR} = 0.016$). Similarly, we find decreased levels of IL1RL1 and IL1RT2, both proteins involved in autoimmunity and inflammation-related disorders[19], to be causally linked to risk of autoinflammatory bowel diseases (Supplementary Data 4).

We further identify new evidence for disease-mediating roles for proteins circulating in the periphery. For example, rs2306272, a missense cis-pQTL, is associated with decreased LRIG1 (leucine rich repeats and immunoglobulin like domains 1) levels (MAF = 31%, meta-analysis $\beta = -0.754$, $\sigma = 0.0261$, $P = 1.50 \times 10^{-183}$), and is causally associated with reduced risk of atrial fibrillation ($P = 5.23 \times 10^{-11}$, $P_{FDR} = 5.32 \times 10^{-8}$) and lower BMI ($P = 2.72 \times 10^{-9}$, $P_{FDR} = 1.89 \times 10^{-6}$), and with increased risk of type 2 diabetes ($P = 4.70 \times 10^{-5}$, $P_{FDR} = 8.87 \times 10^{-3}$) and self-reported

hypercholesterolemia ($P = 4.61 \times 10^{-4}$, $P_{FDR} = 0.043$) (Fig. 4). LRIG1 is a transmembrane protein that acts as a feedback negative regulator of signaling by receptor tyrosine kinases. Variants in *LRIG1* have previously been associated with atrial fibrillation[20,21], and the identified pQTL co-localises with previous pulse rate (posterior probability of colocalisation $P_4 = 0.939$) and QRS duration ($P_4 = 0.998$) loci. Mouse knockout models of *LRIG1* exhibit decreased body weight and fat[22].

Notably, we find evidence for a genetic link between *PRG2* intronic variant rs10642232 and decreased levels of PAPPA (pregnancy-associated plasma protein-A) ($\beta = -0.299$, $\sigma = 0.0320$, $P = 1.06 \times 10^{-20}$). A previous association exists at rs140000161[23], however, this variant is not found in either of our cohorts. PAPPA is a metalloproteinase involved in normal and pathological insulin-like growth factor (IGF) physiology. *PRG2* codes for eosinophil granule major basic protein, which reduces PAPPA activity by interacting with it to form a complex[24]. PAPPA is a specific protease targeting IGFBP4 (IGF binding protein 4) in the presence of IGF. IGFBP4 inhibits IGF binding with its receptor, and PAPPA

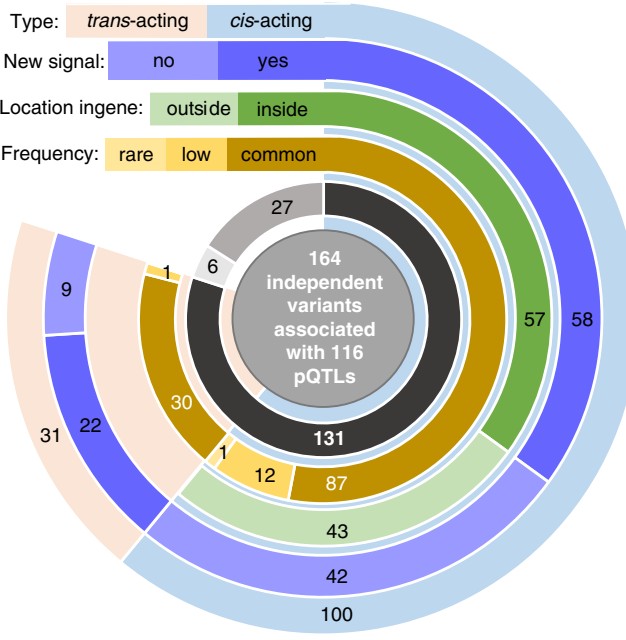

**Fig. 2 Characteristics of independently contributing pQTL variants.** The innermost circle represents replication status: dark grey for variants that replicate, medium grey for variants that do not replicate and light grey for variants for which no proxy was found in the Pomak dataset.

promotes IGF activity[25]. rs10642232 is a *PRG2*-decreasing eQTL in multiple tissues. We find reduced PAPPA levels to be causally associated with decreased risk of diabetic kidney disease in T2D patients ($P = 2.63 \times 10^{-4}$, $P_{FDR} = 0.0304$). IGF activity is enhanced in early diabetic nephropathy, whereas IGF resistance is found in chronic kidney failure[26]. Animal knockouts of PAPPA exhibit decreased body weight and length, type 2 diabetes and hypercholesterolemia. Our results suggest that PAPPA and its inhibition by PRG2 within the IGF system may play a role in the pathogenesis and progression of diabetic kidney disease in T2D patients. These findings are consistent with the reported lower incidence of diabetic complications in this isolated Cretan population[27].

Further, we find that cis-acting variants decreasing levels of ENTPD5 (rs73301485, $MAF = 7.5\%$, $\beta = -0.637$, $\sigma = 0.0487$, $P = 4.59 \times 10^{-39}$; rs140111715, $MAF = 3.7\%$, $\beta = -0.778$, $\sigma = 0.0563$, $P = 2.29 \times 10^{-43}$) are causally associated with lower risk of type 2 diabetes ($P = 3.56 \times 10^{-4}$, $P_{FDR} = 0.037$) and diabetic kidney disease ($P = 2.61 \times 10^{-19}$, $P_{FDR} = 4.93 \times 10^{-16}$). ENTPD5 (ectonucleoside triphosphate diphosphohydrolase 5) promotes glycolysis in proliferating cells in response to phosphoinositide 3-kinase (PI3K) signaling and is primarily expressed in the liver, kidney, intestine, prostate and bladder, and rs73301485 is an eQTL in multiple tissues. Mouse knockout models show decreased body weight, hypoglycemia, decreased cholesterol and triglycerides[28]. Small-molecule screens have recently identified several ENTPD5 inhibitors[29] that warrant investigation for their effect on type 2 diabetes and diabetic complications.

**Polygenic prediction of the cardiometabolic proteome**. We find that genome-wide polygenic scores calculated in MANOLIS can predict up to 45.5% of protein variance in the independent Pomak dataset, despite a low average predictive performance (median $r^2 = 0.026$, Supplementary Fig. 5). The polygenic score architecture observed within the power parameters of this study,

indicates the involvement of a small number of strongly-associated common variants, and a smaller contribution for rare and low-frequency variants. Notably, both the discovery and test datasets stem from individuals of European ancestry; further studies in global populations will be required to assess the transferability of these polygenic scores.

**Predicted protein levels are associated with disease risk**. Polygenic prediction of the cardiometabolic proteome can lead to the identification of potential biomarkers through correlation with disease states in biobanks where clinical and genetic information is available, without requiring actual proteomics measurements. We performed logistic regression of 47 proteins with polygenic scores that had achieved a predictive value of $r^2 > 0.05$ in Pomak, on 80 indications in UK Biobank, adjusted for genetic principal components, clinical and lifestyle factors. We find that the scores for GRN (progranulin), CHI3L1 (chitinase 3 like 1) and PECAM1 (platelet and endothelial cell adhesion molecule 1) levels are significant predictors of disease status (Wald test $P < 1.66 \times 10^{-5}$) across a range of cardiometabolic traits (Supplementary Data 6a, b). The progranulin level polygenic score is correlated with increased risk of hypercholesterolemia, and is driven by a single association in *CELSR2-SORT1*, an established risk locus for lipid disorders[30]. Similarly, the PECAM1 score is driven by a signal at the ABO locus, a known regulator of multiple proteins[4]. In a joint predictive model for high cholesterol, inclusion of polygenic scores for GRN, CHI3L1 and PECAM1 levels results in a significant increase of the model accuracy compared to the clinical and lifestyle covariates-only model (Supplementary Note 2, Likelihood Ratio Test (LRT) $P = 9.07 \times 10^{-126}$, DeLong's test for difference in AUC $P = 3.13 \times 10^{-33}$). This remained significant when GRN and PECAM1 scores were excluded, leaving only the newly associated CHI3L1 score (LRT $P = 7.33 \times 10^{-10}$, DeLong's $P = 3.19 \times 10^{-3}$).We find that an elastic net model agnostically selects the same three polygenic scores in a full-proteome analysis of high-cholesterol, confirming their contribution and demonstrating the value of including proteomics scores in predictive models of disease risk.

In summary, using whole-genome sequencing, we identify robustly-replicating *cis*- and *trans*-pQTLs, and show for the first time that burdens of rare variants contribute to the genetic architecture of protein biomarker levels. We show that incorporating information on this genetic contribution leads to improvement in clinical risk models for cardiovascular disease. Identification of causal contributions of the cardiometabolic proteome to the risk of multiple chronic diseases can present opportunities for new therapeutic target discovery and predictive modeling to accelerate precision medicine.

## Methods
**Sequencing and variant calling**. Genomic DNA (500 ng) from 1482 samples was subjected to standard Illumina paired-end DNA library construction. Adapter-ligated libraries were amplified by 6 cycles of PCR and subjected to DNA sequencing using the HiSeqX platform (Illumina) according to manufacturer's instructions.

Basecall files for each lane were transformed into unmapped BAMs using Illumina2BAM, marking adaptor contamination and decoding barcodes for removal into BAM tags. PhiX control reads were mapped using BWA Backtrack and were used to remove spatial artefacts. Reads were converted to FASTQ and aligned using BWA MEM 0.7.8 to the hg38 reference (GRCh38) with decoys (HS38DH). The alignment was then merged into the master sample BAM file using Illumina2BAM MergeAlign. PCR and optical duplicates are marked using biobambam markduplicates and the files were archived in CRAM format.

Per-lane CRAMs were retrieved and reads pooled on a per-sample basis across all lanes to produce library CRAMs; these were each divided in 200 chunks for parallelism. GVCFs were generated using HaplotypeCaller v.3.5 from the Genome Analysis Toolkit (GATK) for each chunk. All chunks were then merged at sample level, samples were then further combined in batches of 150 samples using GATK CombineGVCFs v.3.5. Variant calling was then performed on each batch using

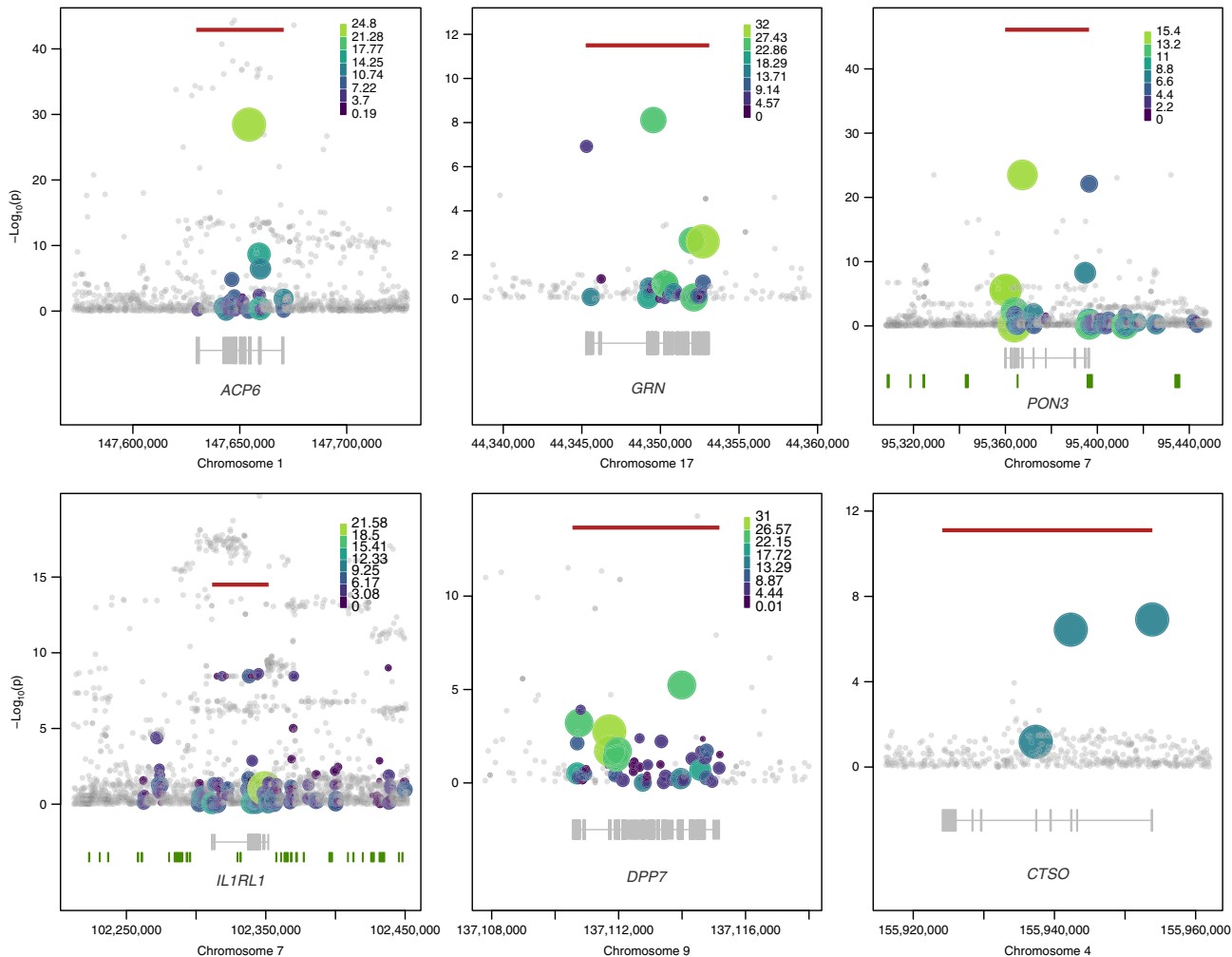

**Fig. 3 Rare variant pQTLs.** Rare variant burden signals detected in this study -the most significant burden per gene is displayed. Circles denote the sequence variants identified in the region. Genes are denoted in gray below the regional association plots; bars represent exons across all transcripts. Horizontal red lines indicate the −log10 of the burden signal p-value, with size and colour of circles proportional to the weighting scheme used (CADD for *ACP6*, *GRN* and *DPP7*, or Eigen for *PON3* and *IL1RL1*. For *CTSO*, where only severe variants are considered, all variants have weights equal to 1. Grey circles denote variants not included in the burden. Details on variants included in each burden are given in Supplementary Data 14.

GATK GenotypeGVCFs v.3.5. The resulting variant callsets were then merged across all batches into a cohort-wide VCF file using bcftools concat.

Variant-level QC was performed using the Variant Quality Score Recalibration tool (VQSR) from the Genome Analysis Toolkit (GATK) v. 3.5-0-g36282e4[31], using a tranche threshold of 99.4% for SNPs, which provided an estimate false-positive rate of 6%, and a true positive rate of 95%. For INDELs, we used the recommended threshold of 1%. For sample-level QC, we made extensive use of a previously described[10] GWAS dataset in 1175 overlapping samples. Four individuals failed sex checks, 8 samples had low concordance ($\pi^\wedge < 0.8\pi^\wedge < 0.8$) with chip data, 11 samples were duplicates, and 12 samples displayed traces of contamination (Freemix score from the verifyBamID suite[32] >5%). In case of sample duplicates, the sample with highest quality metrics (depth, freemix and chipmix score) was kept. As contamination and sex mismatches were correlated, a total of 25 individuals were excluded ($n = 1457$). No further samples were excluded based on depth, heterozygosity, transition/transversion (Ti/Tv) rate, missingness or ethnicity. We filtered out 14% of variants with call rates < 99%.

**Proteomics**. The serum levels of 275 unique proteins in 1407 MANOLIS samples from three Olink panels—CVDII, CVDIII and Metabolism—were measured using Olink's proximity extension assay (PEA) technology[9]. Briefly, for each assay, the binding of a unique pair of oligonucleotide-labelled antibody probes to the protein of interest results in the hybridisation of the complementary oligonucleotides, which triggers extension of by DNA polymerase. DNA barcodes unique to each protein are then amplified and quantified using microfluidic real-time qPCR. Measurements were given in a natural logarithmic scale in Normalised Protein eXpression (NPX) levels, a relative quantification unit. NPX is derived by first adjusting the qPCR $C_t$ values by an extension control, followed by an inter-plate

control and a correction factor predetermined by a negative control signal. This is followed by intensity normalisation, where values for each assay are centered around its median across plates to adjust for inter-plate technical variation. Further details on the internal and external controls used can be found at http://www.olink.com. Additionally, a lower limit of detection (LOD) value is determined for each protein based on the negative control signal plus three standard deviations. For our samples, NPX values that fall below the LOD were set to missing.

We adjusted all phenotypes using a linear regression for age, age squared, sex, plate number, and per-sample mean NPX value across all assays, followed by inverse-normal transformation of the residuals. We also adjusted for season, given the observed annual variability of some circulating protein levels. Given the dry Mediterranean climate of Crete, we define season of collection as hot summer or mild winter. Plate effects are partially offset by the median-centering implemented by Olink. MANOLIS samples were plated in the order of sample collection, which results in plate and season information to be largely correlated.

**Quality control**. We excluded 13 protein measurements across all panels with missingness or below-LOD proportion greater than 40% (Supplementary Data 7). BNP was measured across all three panels, and was excluded due to high missingness in all three. 26, 2, and 14 samples failed vendor QC and were excluded from CVDII, III and META, respectively. 42 samples were excluded due to missing age.

Sequencing data quality control has been described before[7]. Briefly, Variant-level QC was performed using the Variant Quality Score Recalibration tool (VQSR) from the Genome Analysis Toolkit (GATK) v. 3.5-0-g36282e4. Sample-level QC was performed by comparing genotypes with chip data in the same samples. Four individuals failed sex checks, 8 samples had low concordance with chip data,

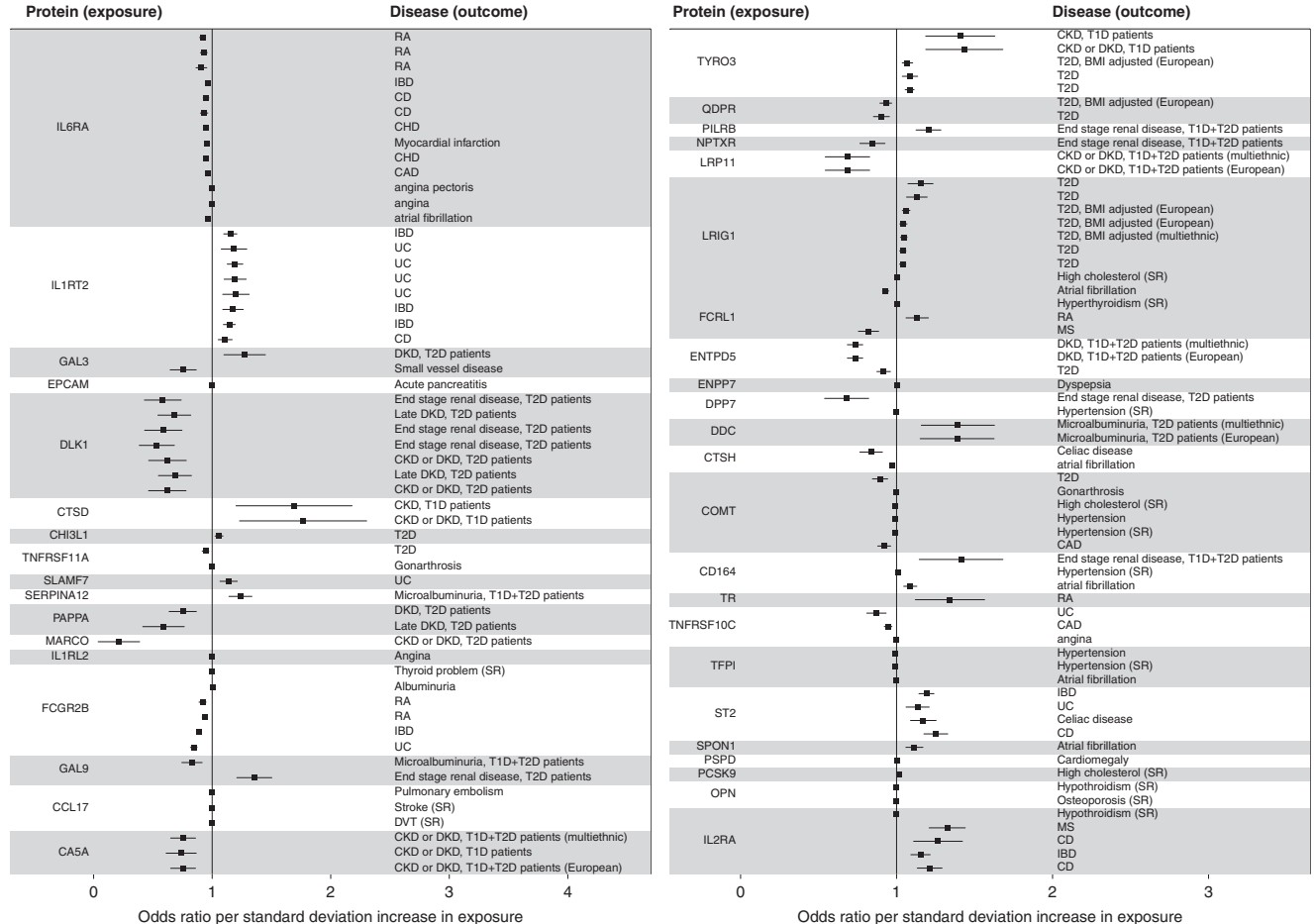

**Fig. 4 Significant causal protein-disease associations identified through two-sample Mendelian randomisation.** Protein (exposure) names are indicated on the left, diseases (outcomes) on the right. Identical disease names for a given protein indicate a MR signal replicating across multiple studies of the same disease; further details and causal associations with quantitative traits are displayed in Supplementary Data 4. RA: rheumatoid arthritis, IBD: inflammatory bowel disease, CD: Crohn's disease, CHD: Coronary heart disease, CAD: Coronary artery disease, UC: ulcerative colitis, DKD: diabetic kidney disease, T2D: type 2 diabetes, T1D: type 1 diabetes, CKD: chronic kidney disease, MS: multiple sclerosis. Error bars denote standard errors.

11 samples were duplicates, and 12 samples displayed traces of contamination. As contamination and sex mismatches were correlated, a total of 25 individuals were excluded ($n = 1457$). Variants were further filtered using the Hardy–Weinberg equilibrium test at $P = 1.0 \times 10^{-5}$. We filtered out 14% of variants with call rates < 99%.

**Single-point association**. We carry out single-point association using the linear mixed model implemented in GEMMA[33]. We use an empirical relatedness matrix calculated on a LD-pruned set of low-frequency and common variants (MAF > 1%) that pass the Hardy–Weinberg equilibrium test ($P < 1 \times 10^{-5}$). We further filter out variants with missingness higher than 1% and MAC < 10. 5 proteins were excluded due to having a genomic control $\lambda_{GC} < 0.97$ or $\lambda_{GC} > 1.05$ after association (total analysed 257, Supplementary Data 7). 123 signals were extracted using the Peak-Plotter software (https://github.com/hmgu-itg/peakplotter), which is based on a combination of distance-based and LD-based pruning. Specifically, the software sorts variants passing the significance threshold by increasing p-value, then for each variant, computes SNPs in linkage disequilibrium greater than $r^2 = 0.2$, removes them, and moves on to the next variant. Variants selected in this way located within less than 2 Mb of each other are then grouped together, and the index variant is set to the variant with lowest $p$-value. Each index variant defines a signal, and we use locus and signal interchangeably in this article. We extracted independent SNV at each associated locus using an approximate conditional and joint stepwise model selection analysis as implemented in GCTA-COJO[34]. To avoid overfitting when too many predictors are included in the model, we perform LD-based clumping using Plink v.1.9, based on an $r^2$ value of 0.1 and a window of 1 Mb prior to the GCTA-COJO analysis[35]. The extended linkage disequilibrium (LD) present within MANOLIS can cause very large peaks to be broken up into several signals. We identified and manually investigated 5 regions where multiple peaks were present in close proximity of each other, reducing the number of independent signals to 116 and the number of conditionally independent variants to 164. Cis-acting protein-altering variants may result in false-positive associations

due to epitope effects. While exact quantification of such effects would have required comparison using proteomic measurements from an alternative assay method, we note that only 11 of these 100 replicating cis-acting variants have a potentially protein-truncating effect. Details of additional notable pQTL not described in the main text are summarised in Supplementary Note 3.

**Rare variant association**. For rare variant association, we apply a MAF filter of 5% and a missingness filter of 1%. We use the linear mixed model extension of SKAT-O implemented in MONSTER[36], using the MUMMY wrapper[7] (https://github.com/hmgu-itg/burden_testing). Following our previously-reported analysis strategy[7], we test for rare variant burden association on a gene-by-gene basis: firstly, restricting burdens to coding variants with Ensembl most severe consequence stronger than missense; secondly, including all coding variants weighted by CADD[37]; thirdly, including exon and regulatory variants using the phred-scaled Eigen score[38]; and, finally, regulatory variants only weighted by Eigen. CADD integrates multiple annotations into one metric by contrasting variants that survived natural selection with simulated mutations, whereas Eigen is an unsupervised method based on spectral decomposition of multiple functional annotations in coding as well as noncoding regions. Regulatory regions are linked to a gene if they overlap an Ensembl-documented eQTL for that gene in any tissue. A gene-pair was taken forward for quality control (QC) if it was significant in any of the four analyses. 17 signals passed this threshold. Because rare variant LD blocks can extend over long distances and capture overlapping common associations, we manually inspect LD blocks through the plotburden software (https://github.com/hmgu-itg/plotburden), and discard signals involving variants in LD with nearby cis ones. For each remaining signal, we then re-run the burden analysis conditional on the genotypes of the variant with the lowest single-point p-value that was previously included in the burden, so as to only consider signals arising from at least two distinct variants. 6 RV-pQTL signals pass this quality control procedure (Supplementary Fig. 4).

**Replication**. We performed replication in 1605 samples from the Pomak cohort sequenced at a mean depth of 18.6×, using an identical sequencing, variant calling and quality control protocol. The proteomic phenotypes were transformed identically to MANOLIS. Single-point association analysis was performed using GEMMA and an identically calculated GRM. The replication p-value was defined as $P_{replication} = 0.05/N_{variants}$, where $N_{variants} = 158$ referred to the total number of independent SNVs associated in MANOLIS and also present in the Pomak cohort. For burden replication, we specifically analysed the genes associated in MANOLIS in all conditions, and defined replication if a significant signal was detected in any of them. The COMT signal driven by rs4680 in MANOLIS ($\beta = -0.373$, $\sigma = 0.0405$, $P = 3.5 \times 10^{-20}$) could not be replicated due to COMT failing QC in Pomak. In five cases, associated MANOLIS variants and those tagged ($r^2 > 0.8$) by them were monomorphic in the Pomak cohort (rs183455943, rs4778724, rs1053361963, rs200251994, rs186044494). 101/116 (87%) loci had at least one replicating variant.

**Definition of novelty**. To assess whether a protein had been previously studied, we examined protein lists and summary statistics from five large published proteomics GWAS[2–4,6,23]. To determine novelty of genetic cis and trans association with proteins in our study, we first determined previously reported variants within a 2 Mb window around the association peaks. Among these five GWAS, one[4] performed stepwise conditional analysis to identify independent variants at associated loci, and three[2,6,23] did LD-based detection of independent signals. We were unable to perform independent variant detection for the remaining study[3] since no summary statistics were publicly available. We used GEMMA[33] to perform association analysis using previously-reported independent variants as covariates. The association signals were declared novel if either there were no known signals in the 2 Mb window, or the associations were still study-wide significant ($P$-value threshold: $7.45 \times 10^{-11}$) after conditioning. For trans associations, we further annotated signals depending on whether they fell within highly pleiotropic genes that were associated with more than 1 protein in the current study and had evidence of additional associations in the literature (*KLKB1, ABO, APOE, FUT2, F12*), or whether they were independent of any cis signals in the vicinity. After this procedure, 58 cis-associated variants in 44 loci were either not within 1 Mb or independent of a signal reported in previous proteomics GWAS. 22 trans-associated variants were both novel and independent from cis loci. 11 of these were not located within highly pleiotropic genes. For all loci annotated as provisionally novel using the above method, we queried the GWAS Catalog through the Ensembl REST API[39], as well as PhenoScanner[39] in a 2 Mb window around the lead SNP. Since proteomics GWAS signals are often designated generically in Ensembl, we additionally performed direct queries to the GWAS catalog REST API when phenotype descriptions were not specific enough. We manually investigated the list of signals in search of variants associated with the protein trait of interest. When such a variant was found, conditional analysis was performed and the novelty status updated accordingly. We further incorporated evidence from three association studies with signals that were not reported in the GWAS catalog[40–42]. Using this method, there were thirty-nine proteins measured in this study for which we were not able to find evidence of previous studies of genetic associations. We find 8 cis loci harbouring 11 independent associated variants and one trans locus for 9 of these proteins.

**Variant consequences**. Consequence was evaluated using Ensembl VEP[43] for each variant with respect to any transcript of the cis gene for cis-associated variants and to the mapped gene for trans-associated variants. For trans associations, variants were manually mapped to any gene in a 1 Mb window coding for known ligands or interactants when they were not contained within gene boundaries, as was the case for CXCL16 and LDLR. In all, 16 replicating independent variants had a most severe consequence equal to or more severe than missense according to Ensembl VEP. For every variant, we extracted tagging SNVs at $r^2 > 0.8$ using PLINK, however none of these tagging variants had a more severe consequence on the target gene than the independent variant. Similarly, we overlapped all independent variants with regulatory features using the Ensembl REST API. 35 variants in 29 loci overlapped with a regulatory feature. When extending the same analysis to variants with $r^2 > 0.8$ with independent variants, 93 variants in 68 loci could be mapped to a regulatory feature. We further examined potential variant consequences by performing eQTL overlap analysis, as well as mining of drug targets and mouse models databases. The details are provided in the Supplementary Note 4.

**Two-Sample MR**. We extracted variants characterized as independent signals by GCTA-COJO on a protein-by-protein basis across all cis and trans loci, and excluded novel variants without an rs-ID. For each remaining variant, we then considered summary statistics for all tagging positions with $r^2 > 0.8$, merged the resulting data frame with the exposure dataset by rs-ID. All such records originating from all independent signals were then merged by protein and carried over to MR analysis using the MRBase R package. We excluded trans pleiotropic loci (*ABO, KLKB1, FUT2, APOE, F12*). MR was performed on a set of 127 medically-relevant traits available in MRBase (Supplementary Data 3a). Since all of our instruments involved a small number of variants (≤10), we used the inverse-variant

weighted method, except for single-instrument analyses where we use the Wald ratio test, which consists of dividing the instrument-outcome by the instrument-exposure regression coefficient. We note that all of the GWAS summary statistics used in this analysis were not derived from WGS-based studies, and therefore several of our instruments were not found in these datasets and could not be used. An important caveat of our overlap-maximising approach is that we did not require overlapping variants to be lead variants in the outcome trait GWAS. This could potentially lead to false-positives for single-instrument tests if the variant is located at the shoulders of an association peak in the outcome trait GWAS. The future availability of population-scale association studies with WGS or WES will greatly enhance variant overlap compared to GWAS, and hence increase the power of MR analyses in proteomics.

We also leveraged summary statistics manually downloaded from recent large association studies for Chronic Kidney Disease[44], blood lipids[45], Atrial Fibrillation, Type-II Diabetes, Coronary Artery Disease, estimated glomerular filtration rate[46], albuminuria[47], and anthropometric traits[48].

Two proteins, PDL2 and TNFRSF10C, had both cis and trans associations where the trans did not fall in a highly pleiotropic gene. We performed 2-sample MR excluding trans, and found that no cis variant in TNFRSF10C could be found in selected external studies. This was due to the lead variant, 8:23108277T/TG, being novel, and secondary variants, such as rs779159813, being very rare (MAF = 0.573%). For PDL2, the known cis signal driven by rs62556120 ($\beta = 0.416$, $\sigma = 0.0276$, $P = 2.92 \times 10^{-51}$) was causally associated with increased risk of ulcerative colitis and inflammatory bowel disease (Supplementary Data 8), whereas in the cis+trans analysis, the addition of rs10935473 attenuates that signal, but drives a causal association with height.

We note that due to the small number of instruments for nearly all (189/13,207, 1.4%) protein-outcome pairs, we were unable to apply analyses that account for violations of MR assumptions, such as MR-Egger regression. As such, we were unable to assess horizontal pleiotropy, in particular for trans effects, within the power constraints of this study. In the future, larger meta-analyses will be needed to produce a higher number of independent and robust instruments, enabling a more nuanced analysis of the causal relationships between proteins and disease.

**Significance thresholds**. We calculate the significance thresholds by computing the effective number of variants, traits and analyses for every analysis requiring multiple-testing correction.

Single-variant analyses: The effective number of proteins was computed using the ratio of the eigenvalue variance to its maximum[49,50]:

$$M_{eff} = M\left(1 - (M-1)V_{\lambda_{obs}}/M^2\right) = 1 + \frac{tr\left(\sum^{T}\sum\right)}{M}$$

where $V_{\lambda_{obs}}$ is the variance of the eigenvalues of the correlation matrix. For the $M = 257$ Olink phenotypes in this study, $M_{eff} = 131.5$, which we round to 132. The resulting p-value threshold is $7.45 \times 10^{-11}$.

Rare-variant analyses: We report both single-point and rare variant burden signals, therefore increasing the multiple testing burden. The exact magnitude of this phenomenon being unknown, we performed a simulation study to compute the effective number of tests in case single-variant, variant-aggregation or both are reported in an association study. We find that reporting rare variant signals in combination with single-point signals at 5% and 1% MAF thresholds increased the multiple testing burden only marginally and by less than one order of magnitude (Supplementary Fig. 6).

Two-sample MR: To correct for multiple testing in our MR analysis, we adjust p-values using FDR correction and examine significant results at an FDR of 0.05.

Polygenic prediction: To examine predictive accuracy, we compute polygenic scores using PRSice 2[51] (v 2.2.6) with the Pomak as a target dataset. This software applies LD-pruning and p-value thresholding on the input variant, and subsequently performs optimization of the p-value threshold against prediction accuracy, as measured by R-squared, in an external validation cohort. Using Pomak as the validation cohort, we assess scores between $1 \times 10^{-4}$ and study-wide significance with intervals of $1 \times 10^{-10}$. We further apply three MAF thresholds, at 0.05, 0.01 and MAC = 10, which produces three best scores per protein. We find allele frequency thresholds not to have an appreciable influence on predictive power. The scores that achieve high accuracy ($r^2 > 0.05$) in predicting Pomak protein levels all involve stringent thresholds ($P < 1 \times 10^{-6}$, with the majority at $P < 1 \times 10^{-9}$, Supplementary Data 9). We examine correlation of these scores and find five risk scores (CTRC, SELE, ICAM2, CDH5, PECAM1) to be highly correlated due to signals present in the *ABO* region. After exclusion of a 2 Mb window centred around the *ABO* gene, the ICAM2 score is the only one that maintains $r^2 > 0.05$ in Pomak (Supplementary Data 10c). To guard against potential overfitting, we perform three rounds of repeated 5-fold cross-validation with a 20% hold-out set, and obtain similar results to the full analysis (Supplementary Data 10a,b), with consistent ranking of the 7 top proteins in terms of $r^2$. To evaluate the predictive power of proteins for complex disease, we leverage medical and genetic information available in the UK Biobank. Since proteomic measurements are not available in that cohort, we use polygenic scores to impute protein levels and correlate them with disease. We compute all scores between $1 \times 10^{-6}$ and study-wide significance with the same interval. We then run a logistic regression of 80 UK Biobank self-reported disease codes (Supplementary Data 11a)

and ICD codes (Supplementary Data 11b) on all such scores on a protein-by-protein basis for 47 proteins that achieved $r^2 > 0.05$ in the previous step. Sex, age, Qualification, Smoking status and BMI as well as 10 principal components are also added to the model. In order to accurately assess effect sizes, we select the most predictive score for each protein where at least one score threshold meets the Bonferroni-corrected P-value threshold (47 proteins, 64 effective phenotypes, $P < 1.66 \times 10^{-5}$) in the Wald test for variable contribution. We then run the same model as before, with only this score and covariates as predictors. We removed associations between 3 proteins (CTRC, ICAM2, SELE) and deep vein thrombosis and/or pulmonary embolism due to the association being entirely driven by *ABO* variants.

In order to validate our approach for variable selection, we pooled all scores for all proteins in a single model of self-reported high cholesterol, and used elastic net regression to shrink coefficients for non-informative predictors. We run 10-fold cross-validation with a 20% hold-out sample to optimize the lambda value, and run 11 models using different values of alpha, from 0 (ridge regression) to 1 (lasso) at intervals of 0.1. We use a value of lambda one standard deviation away from the minimal value to increase shrinkage. Overall, the performance was comparable across models, but small values of alpha yielded a better predictive performance on the hold-out set. An alpha value of 0.1 yielded almost no loss of AUC compared to ridge regression, but shrunk almost all coefficients to 0 (Supplementary Fig. 5). Four protein scores (CHI3L1, PECAM1, SELE and GRN) were shrunk to a value greater than 0, confirming results of the manual variable selection procedure.

A disease may be manifested as several ICD codes, and broadening phenotype definition beyond single ICD codes could increase case numbers and thereby boost power. We use the PheCode Map 1.2 with ICD-10 codes[52] to first select any PheCode whose definition included at least one ICD-10 code included in our previous analysis. We then used the union of cases for all ICD-10 codes included in those PheCode definitions to define case/control status, and repeated the analysis. Regression outputs are presented in Supplementary Data 12 and confirm the single-protein results obtained with ICD-10 codes. A discussion of the additional protein signals discovered using the PheCode analysis is detailed in the Supplementary Note 5.

Finally, related individuals are present in UK Biobank, which might result in test statistic inflation when predicting disease risk. We perform a sensitivity analysis excluding related individuals, which confirms the robustness of our multi-protein models (Supplementary Note 6).

**Reporting summary**. Further information on research design is available in the Nature Research Reporting Summary linked to this article.

## Data availability

Sequencing data are available at the European Genome-Phenome Archive (EGA) under accession number EGAS00001001207 for MANOLIS. The Pomak sequencing data have not been deposited to the EGA due to the sensitive nature of this population. They are available through email request from the corresponding author, for research projects not involving contentious topics such as population genetics. Summary statistics are available through the GWAS catalog. Accession numbers are provided in Supplementary Data 13. Summary statistics for MR, including UK Biobank, were accessed using the MRBase R package. Download links for publicly available datasets not included in MRBase are available in Supplementary Data 3b. We leveraged information from the DrugBank[53], OpenTargets[54], Ensembl release 100[55], and IMPC[56] databases for variant and gene annotation. Source data are provided with this paper.

## Code availability

Analysis was performed using publicly available software as described in the Methods. We have deposited scripts on GitHub (github.com/hmgu-itg).

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

## Acknowledgements

We thank the residents of the Pomak and Mylopotamos villages for taking part. The MANOLIS study is dedicated to the memory of Manolis Giannakakis, 1978–2010. This work was funded by the Wellcome Trust [098051] and the European Research Council [ERC-2011-StG 280559-SEPI]. The GATK3 program was made available through the generosity of the Medical and Population Genetics program at the Broad Institute, Inc. We thank the Human Genetics DNA Pipelines and Human Genetics Informatics departments at the Wellcome Sanger Institute for performing sequencing and variant calling. This study has been conducted using the UK Biobank Resource (project ID 10205).

## Author contributions

E.T., M.K., G.D., and E.Z. designed and carried out sample collection and phenotyping. A.G., Y.C.P., G.P., L.S., and N.W.R. performed phenotype transformation and quality control. The association analysis was carried out by A.G. and Y.C.P. using software developed by A.G and D.S. Additional bioinformatics analyses were performed by A.G., S.N., I.F., and A.B. The power simulation analysis was implemented by T.B. E.Z. was responsible for study design and supervision, and contributed to manuscript writing along with A.G. and I.F.

## Funding

## Competing interests

The authors declare no competing interests.

## Ethics statement

The study was approved by the Institutional Review Board of Harokopio University and the Greek Ministry of Education, Lifelong Learning and Religious Affairs. The MANOLIS and Pomak studies were approved by the Harokopio University Bioethics Committee and informed consent was obtained from every participant.
