## [Peer Review File · Nature Communications]

Reviewer #1 (Remarks to the Author):

This manuscript describes a study of a protein-GWAS where 257 proteins were assessed in two sequenced cohorts: Manolis (N=1,328) and Pomak (n=1,605). The authors perform a variety of state-of-the-art analyses including GWAS for all proteins (pQTLs), overlap of pQTLs with GWAS catalogue signals, gene-based burden association with protein levels, two-sample MR between pQTLs and GWAS catalogue signals, and finally generating genome-wide polygenic scores for protein levels and testing the PRSs for association in UK biobank for 87 indications.

I really enjoyed reading this paper - it is an interesting approach and lots of interesting biology. My comments are mostly superficial. I thought the statistical analysis was sound and appropriate, and the two-sample MR a particular asset to the manuscript.

I did find the descriptive stats of the pQTLs to get a little number-heavy. e.g. the whole section from "117 (90%) of these 131 reproducibly-associated variants are common (minor allele frequency (MAF) >5%)" to "Of the replicating 31 trans-acting variants, 30 are common and one is low-frequency." to be kinda boring. Is there a way to put this in the supplement or in a table and give a little more biology or something there? I see the relevance of the comparison of rare and common variants, but the rest seemed tedious to read.

The figures, however, were really interesting to look at and I commend you for exceptional data visualization.

Nice work!

Reviewer #2 (Remarks to the Author):

Summary

The manuscript of Arthur et al describes a large scale whole-genome sequencing (WGS) to identify genetic variation associated with protein-level expression (i.e., pQTLs) of n=257 proteins, performed initially in ~1328 individuals (with replication in additional cohorts). They reported 131 independent and replicated sequence variant association, with an additional identification for the first time of rare-variant cis-acting pQTLs for five genes. They then perform two sample Mendelian Randomization(MR) to generate evidence for causal hypotheses to complex traits and construct and validate polygenic scores that explain up to 45% of protein level variation.

Overview.

This is a methodologically sound and thorough analysis and is valuable in gaining insight into the mechanisms of genetic variants associated with protein expression. However, while the high-depth of WGS is significant, the overall contribution - novelty, fell modestly. For example, what's the strength of WGS on pQTL study. What's the value of rare-variant cis-acting pQTLs for five genes? And there are some points that require clarification to aid understanding, such as the organization of the text and the supplementary figures makes it very difficult to follow. Several supplementary figures are either misplaced or misnamed. Major and Minor points are included below:

Major Comments

1. Single-point association.

1a. What transformation of protein expression level is used in the linear mixed model? Log transferred?

1b. The significant P value threshold from cis and trans should be different. In cis, you only test variants within 1Mb of the gene for association, but in trans, it's close to full genome-wide .

2. Colocalisation testing for eQTL overlap and pheWAS analysis

Maybe I missed something, but this part is not in the results section, and the methods section contain both methods and results.

3. Mendelian Randomization

3a. The MR test feels like it needs to be managed a bit more carefully, particularly using both cis and trans pQTLs in the multi-SNP scores. Are the trans effects of SNPs more likely to violate MR assumptions?

3b. How did the author select outcomes? The author stated MR is to detect proteins that may play a causal role in cardiometabolic disease onset or progression, but there are many non-cardiometabolic diseases in 193 outcomes.

3c. The author used FDR for adjusting multiple testing, but in supplementary table 4, the pBH indicated Bonferroni correction.

4. Polygenic prediction

I don't understand use polygenic prediction to identify the potential protein biomarkers. You have measured the protein expression, why not use the protein-disease association to identify the predictors?

5. Supplementary figures

5a. Supplementary Figure 1: cis-association within the CTSB gene didn't match the main text description "complex allelic architecture at pQTLs";

5b. Supplementary Figure 2: Effect size according to log-MAF for independent variants at pQTLs discovered in this study didn't match the main text description "replicating evidence"

5c. Supplementary Figure 6 and 7 are flipped.

Minor Comments

6. Supplementary table 2 is not well formatted.

7. Supplementary table 13a and b are not cited separately in the main text.

8. Supplementary table 15 are not cited in the main text.

9. Supplementary Figure 3: Variance explained in proteomic traits compared with 37 non-proteomic traits. How about comparing pQTLs with eQTLs?

Reviewer #3 (Remarks to the Author):

The authors performed extensive genetic association analyses on 257 proteins of cardiometabolic relevance, and used genetic association findings to establish potentially causal links between the proteins and various diseases. The authors should be commended for their careful work to identify independent association signals/SNPs and differentiate novel vs. existing findings. Their work can be a valuable contribution to the body of literature on genetic architecture of proteome.

Here are some specific comments:

1. The authors need to clarify what they considered loci and signal, and the relationship with lead SNPs in Supplementary table 1. The author need to detail the criteria PeakPlotter used to identify independent signals, and their manual work to further reduce number of signals.

2. In Figure 3, size and colour of circles were proportional to the weighting scheme used (CADD or Eigen). The authors need to clarify which corresponds to CADD and which corresponds to Eigen. In the text, the author need to briefly explain the methods of CADD and Eigen.

3. Please describe replication p-value threshold in the method section where replication analyses were described.

4. Please describe and justify the prior probability for colocalization and what posterior probability threshold is used to declare colocalization. A result table is needed to report protein and gene-expression pairs found colocalized.

5. There is a lack of assessing the validity of MR analysis such as examining horizontal pleiotropy.

6. Please describe the methodology of PRSice 2.

7. A disease may be manifested as several ICD codes. Current analyses with a single ICD code may only partially capture some of the diseased subjects. For example, the following ICD codes are considered hypotension:

I95 Hypotension

I95.0 Idiopathic hypotension

I95.8 Other hypotension

I95.9 Hypotension, unspecified

but the authors only included I95.9 in their analyses. The authors should consider using PheCode

to define disease, such as https://phewascatalog.org/phecodes_icd10

8. The number of cases and controls should be reported for each disease in UKBB samples. There are a lot of related individuals in UKBB samples, the authors didn't mention how those were dealt with. Ignoring relatedness in samples could result in inflation.

9. The authors stated that they evaluate the predictive power of proteins for complex disease. But they performed only association analyses. AUC or R-squared should be used to evaluate prediction power in addition to p-value from association analyses.

REVIEWER COMMENTS

Reviewer #1 (Remarks to the Author):

This manuscript describes a study of a protein-GWAS where 257 proteins were assessed in two sequenced cohorts: Manolis (N=1,328) and Pomak (n=1,605). The authors perform a variety of state-of-the-art analyses including GWAS for all proteins (pQTLs), overlap of pQTLs with GWAS catalogue signals, gene-based burden association with protein levels, two-sample MR between pQTLs and GWAS catalogue signals, and finally generating genome-wide polygenic scores for protein levels and testing the PRSs for association in UK biobank for 87 indications.

I really enjoyed reading this paper - it is an interesting approach and lots of interesting biology. My comments are mostly superficial. I thought the statistical analysis was sound and appropriate, and the two-sample MR a particular asset to the manuscript.

1. I did find the descriptive stats of the pQTLs to get a little number-heavy. e.g. the whole section from "117 (90%) of these 131 reproducibly-associated variants are common (minor allele frequency (MAF) >5%)" to "Of the replicating 31 trans-acting variants, 30 are common and one is low-frequency." to be kinda boring. Is there a way to put this in the supplement or in a table and give a little more biology or something there? I see the relevance of the comparison of rare and common variants, but the rest seemed tedious to read.

We thank the reviewer for their suggestion. We have rewritten the paragraph as follows and fused it with the one immediately below, in order to improve clarity:

"Ninety percent of these reproducibly-associated variants are common (minor allele frequency (MAF) >5%), and 76% are located within 1Mb of the gene encoding the respective protein (i.e. in cis-pQTLs) (Figure 2). Among these cis loci, 32 out of 72 cis-pQTLs (44%) discovered in this cohort have either not previously been reported in protein-level GWAS (novel loci), or harbour variants conditionally independent of all previously-reported associations (novel variants at known loci) (Supplementary Table 1)."

We have moved the numerical details to a new section in the Supplementary Text entitled "**Overview of single-point cis signals**":

"117 (90%) of 131 variants reproducibly associated in the Pomak cohort are common (minor allele frequency (MAF) >5%), 13 are low-frequency (MAF 1-5%) and 1 is rare (MAF<1%) (Figure 2). One hundred of the

associated variants (76%) are located within 1Mb of the gene encoding the respective protein (i.e. in *cis*-pQTLs), and 31 (24%) are in *trans*-pQTLs (Figure 2). 57 *cis*-associated variants are located within the boundaries of the respective gene, and the remaining 43 are at a median distance of 13.8kb (max. 920 kb) (Supplementary Figure 7)."

Reviewer #2 (Remarks to the Author):

Summary.

The manuscript of Arthur et al describes a large scale whole-genome sequencing (WGS) to identify genetic variation associated with protein-level expression (i.e., pQTLs) of n=257 proteins, performed initially in ~1328 individuals (with replication in additional cohorts). They reported 131 independent and replicated sequence variant association, with an additional identification for the first time of rare-variant *cis*-acting pQTLs for five genes. They then perform two sample Mendelian Randomization(MR) to generate evidence for causal hypotheses to complex traits and construct and validate polygenic scores that explain up to 45% of protein level variation.

Overview.

This is a methodologically sound and thorough analysis and is valuable in gaining insight into the mechanisms of genetic variants associated with protein expression. However, while the high-depth of WGS is significant, the overall contribution - novelty, fell modestly. For example, what's the strength of WGS on pQTL study. What's the value of rare-variant *cis*-acting pQTLs for five genes? And there are some points that require clarification to aid understanding, such as the organization of the text and the supplementary figures makes it very difficult to follow. Several supplementary figures are either misplaced or misnamed. Major and Minor points are included below:

Major Comments

1. SINGLE-POINT ASSOCIATION

1a. What transformation of protein expression level is used in the linear mixed model? Log transferred?

These details are provided in the methods section, "Measurements were given in a natural logarithmic scale in Normalised Protein eXpression (NPX) levels, a relative quantification unit. NPX is derived by first adjusting the qPCR Ct values by an extension control, followed by an inter-plate control and a correction factor predetermined by a negative control signal. This is followed by intensity normalisation,

where values for each assay are centered around its median across plates to adjust for inter-plate technical variation.”

We also changed the ambiguous term “normalisation” to “inverse-normal transformation” in the following paragraph:

“We adjusted all phenotypes using a linear regression for age, age squared, sex, plate number, and per-sample mean NPX value across all assays, followed by **inverse-normal transformation** of the residuals. We also adjusted for season, given the observed annual variability of some circulating protein levels. Given the dry Mediterranean climate of Crete, we define season of collection as hot summer or mild winter. Plate effects are partially offset by the median-centering implemented by Olink. MANOLIS samples were plated in the order of sample collection, which results in plate and season information to be largely correlated.”

1b. The significant P value threshold from cis and trans should be different. In cis, you only test variants within 1Mb of the gene for association, but in trans, it’s close to full genome-wide.

We agree with the reviewer that in a two-tiered analysis, first checking for cis-effects in +/-1Mb, and then for trans in the rest of the genome, different thresholds can be used. While this is a valid approach that is sometimes used in proteomics studies, and while it does indeed produce two thresholds, this is not the testing framework we used in this work. Instead, we chose to test genome-wide for all proteins, and classify signals into cis and trans a posteriori. We have therefore applied a single, more stringent threshold.

2. COLOCALISATION TESTING FOR EQTL OVERLAP AND PHEWAS ANALYSIS

Maybe I missed something, but this part is not in the results section, and the methods section contain both methods and results.

We thank the reviewer for this comment. We have moved the sections “Colocalisation testing for eQTL overlap and pheWAS analysis”, “Drug Target evaluation”, and “Mouse models” to the Supplementary Text in order to improve readability. We have also added the following sentences to reference these paragraphs at the end of the “**Variant consequences**” section: “We further examined potential variant consequences by performing eQTL overlap analysis, as well as mining of mouse models and drug target databases. The details are provided in the Supplementary Text.”

3. MENDELIAN RANDOMIZATION

3a. The MR test feels like it needs to be manage a bit more carefully, particularly using both cis and trans pQTLs in the multi-SNP scores . Are the trans effects of SNPs more likely to violate MR assumptions?

We thank the reviewer for pointing out this limitation. Including both *cis* and *trans* effects is indeed more likely to produce violations of MR assumptions. This is relevant for two proteins tested in the MR framework here, as explicitly mentioned in the methods section: “Two proteins, *PDL2* and *TNFRSF10C*, had both *cis* and *trans* associations where the *trans* did not fall in a highly pleiotropic gene. We performed 2-sample MR excluding *trans*, and found that no *cis* variant in *TNFRSF10C* could be found in selected external studies. This was due to the lead variant, 8:23108277 T/TG, being novel, and secondary variants, such as rs779159813, being very rare (MAF= 0.573%). For *PDL2*, the known *cis* signal driven by rs62556120 ($\beta=0.416$, $\sigma=0.0276$, $P=2.92 \times 10^{-51}$) was causally associated with increased risk of ulcerative colitis and inflammatory bowel disease (Supplementary Table 8), whereas in the *cis+trans* analysis, the addition of rs10935473 attenuates that signal, but drives a causal association with height.” As this section mentions, even though *cis* and *trans* loci are included in the multi-SNP score for *TNFRSF10C*, none of the *cis* variants overlap, and the MR analysis reduces to a *trans*-only analysis for that protein.

We had considered employing analyses that account for violations of MR assumptions, e.g. the Egger test. However, MR-Egger requires the inclusion of at least three, and ideally more, un- or weakly-correlated variants in the regression, which would only have been possible for 189/13,207 (1.4%) protein/outcome combinations. We have added a cautionary note in the text: “We note that due to the small number of instruments for nearly all protein-outcome pairs, we were unable to apply analyses that account for violations of MR assumptions, such as MR-Egger regression. As such, we were unable to assess horizontal pleiotropy, in particular for *trans* effects, within the power constraints of this study. In the future, larger meta-analyses will be needed to produce a higher number of independent and robust instruments, enabling a more nuanced analysis of the causal relationships between proteins and disease.”

3b. How did the author select outcomes? The author stated MR is to detect proteins that may play a causal role in cardiometabolic disease onset or progression, but there are many non-cardiometabolic disease in 193 outcomes.

We thank the reviewer for pointing out this inconsistency. We have changed “cardiometabolic” to “medically-relevant” in the section that describes the manual selection of outcomes.

3c. The author used FDR for adjusting multiple testing, but in supplementary table 4, the pBH indicated Bonferroni correction.

We thank the reviewer for pointing this out. In Supplementary Table 4, pBH actually refers to the Benjamini-Hochberg corrected p-value, as indicated in the legend. This correction was computed using the

p.adjust function in the R package 'stats' using the 'BH' or 'fdr' argument. This is now clearly stated in the table legend.

4. POLYGENIC PREDICTION

I don't understand use polygenic prediction to identify the potential protein biomarkers. You have measured the protein expression, why not use the protein-disease association to identify the predictors?

We perform PRS estimation in HELIC MANOLIS, validate the scores in HELIC Pomak, and use them to perform prediction in UK Biobank. In the HELIC cohorts, protein measurements are available, but not biobank-quality clinical data. Furthermore, the relatively much smaller sample size of the HELIC cohort would not be suitable for performing (cross-)validation of a polygenic model of disease. In contrast, UK Biobank has detailed information about diseases and clinical records, but no protein measurements. We impute protein levels in UK Biobank using scores computed in HELIC and correlate those with disease status. In order to make this clearer, we changed the first sentence of the polygenic prediction paragraph as follows:

"Polygenic prediction of the cardiometabolic proteome can lead to the identification of potential biomarkers through correlation with disease states in biobanks where clinical and genetic information is available, without requiring actual proteomics measurements."

We also added the following sentence to the corresponding Methods section:

"To evaluate the predictive power of proteins for complex disease, we leverage medical and genetic information available in the UK Biobank. Since proteomic measurements are not available in that cohort, we use polygenic scores to impute protein levels and correlate them with disease. "

5. SUPPLEMENTARY FIGURES

5a. Supplementary Figure 1: cis-association within the CTSH gene didn't match the main text description" complex allelic architecture at pQTLs";

Thank you for pointing this out. We have moved the reference to earlier in the text: *"Thirty-two (27%) of these are driven by multiple independent variants (between two and seven per locus (Supplementary Figure 1), giving rise to a total of 164 independently-associated variants) illustrating complex allelic architecture at pQTLs."*

5b. Supplementary Figure 2: Effect size according to log-MAF for independent variants at pQTLs discovered in this study didn't match the main text description "replicating evidence"

We have made the reference clearer by splitting the original sentence in two and adding a clarification as to what SF2 depicts: *"We find*

replicating evidence for association ($P < 0.000305$) across 131 out of 159 variants (82%) present in an independent, whole genome-sequenced population-based cohort with the same serum biomarker measurements (Pomak) ($n = 1,605$, 18.4x WGS). Replication was expectedly poorer for rare variants (Supplementary Figure 2)."

5c. Supplementary Figure 6 and 7 are flipped.

The figures have been reordered (are now Supplementary Figure 5 and 6).

Minor Comments

6. Supplementary table 2 is not well formatted.

Thank you, the table has now been parsed into an Excel table.

7. Supplementary table 13a and b are not cited separately in the main text.

We now reference both subtables separately (now ST11): *"We then run a logistic regression of 80 UK Biobank self-reported disease codes (Supplementary Table 11a) and ICD codes (Supplementary Table 11b) on all such scores on a protein-by-protein basis for 47 proteins that achieved $r^2 > 0.5$ in the previous step."*

8. Supplementary table 15 are not cited in the main text.

The Supplementary Table (now ST14) is referenced by Supplementary Figure 3.

9. Supplementary Figure 3: Variance explained in proteomic traits compared with 37 non-proteomic traits. How about comparing pQTLs with eQTLs?

In this analysis, we sought to compare the levels of variance explained in the same cohort of individuals across different types of traits. While we agree that a comparison with eQTL effects would be of interest, we do not have eQTL information in the HELIC cohorts.

Reviewer #3 (Remarks to the Author):

1. The authors need to clarify what they considered loci and signal, and the relationship with lead SNPs in Supplementary table 1. The author need to detail the criteria PeakPlotter used to identify independent signals, and their manual work to further reduce number of signals.

We thank the reviewer for this comment and have now amended the methods to clarify our nomenclature and to explain how PeakPlotter works: *"Specifically, the software sorts variants passing the significance*

threshold by increasing p-value, then for each variant, computes SNPs in linkage disequilibrium greater than $r^2=0.2$, removes them, and moves on to the next variant. Variants selected in this way located within less than 2Mb of each other are then grouped together, and the index variant is set to the variant with lowest p-value. Each index variant defines a signal, and we use locus and signal interchangeably in this article." As stated in the methods, computation of independent signals is not performed using PeakPlotter, but using COJO. The manual procedure is described in the following sentence, which we have amended for clarity: "The extended linkage disequilibrium (LD) present within MANOLIS can cause very large peaks to be broken up into several signals. We identified and manually investigated 5 regions where multiple peaks were present in close proximity to each other, reducing the number of independent signals to 116 and the number of conditionally independent variants to 164. "

2. In Figure 3, size and colour of circles were proportional to the weighting scheme used (CADD or Eigen). The authors need to clarify which corresponds to CADD and which corresponds to Eigen.

In the text, the author need to briefly explain the methods of CADD and Eigen.

We have now updated the legend of Figure 3 as follows: "Horizontal red lines indicate the $-\log_{10}$ of the burden signal p-value, with size and colour of circles proportional to the weighting scheme used (CADD for a. b. and e. or Eigen for c. and d.). In f. where only severe variants are considered, all variants have weights equal to 1."

Furthermore, we have added the following sentences to the rare variant testing section of the methods:

"CADD integrates multiple annotations into one metric by contrasting variants that survived natural selection with simulated mutations, whereas Eigen is an unsupervised method based on spectral decomposition of multiple functional annotations in coding as well as noncoding regions."

3. Please describe replication p-value threshold in the method section where replication analyses were described.

We have now made this clear in the Replication section of the methods: "The replication p-value was defined as $P_{\text{replication}}=0.05/N_{\text{variants}}$, where $N_{\text{variants}}=158$ referred to the total number of independent SNVs associated in MANOLIS and also present in the Pomak cohort."

4. Please describe and justify the prior probability for colocalization and what posterior probability threshold is used to declare colocalization. A result table is needed to report protein and gene-expression pairs found colocalized.

We have expanded the corresponding methods section, now located in the Supplementary data, to detail priors and posterior thresholds used in the colocalization analysis:

"We use the commonly chosen value of 0.8 as a posterior threshold to declare colocalization (Tachmazidou et al. 2019), and default values of 1×10^{-4} , with standard deviation of 1, for the prior probability of a variant to be causal for either trait, and 1×10^{-5} , with standard deviation of 1, for the prior probability of a variant to be causal for both traits."

The full colocalization results are presented in Supplementary Table 15 as cited in the text: *"82% of all signals co-localised with previous phenotypic associations (Supplementary Table 15)."*

5. There is a lack of assessing the validity of MR analysis such as examining horizontal pleiotropy.

We agree with the reviewer. However, application of tests for horizontal pleiotropy were not possible due to the small number of instruments. Please also see our response to Reviewer 2, question 3.a.

6. Please describe the methodology of PRSice 2.

We have expanded the method section as follows: *"This software applies LD-pruning and p-value thresholding on the input variant, and subsequently performs optimization of the p-value threshold against prediction accuracy, as measured by R-squared, in an external validation cohort."*

7. A disease may be manifested as several ICD codes. Current analyses with a single ICD code may only partially capture some of the diseased subjects. For example, the following ICD codes are considered hypotension:

I95 Hypotension

I95.0 Idiopathic hypotension

I95.8 Other hypotension

I95.9 Hypotension, unspecified

but the authors only included I95.9 in their analyses. The authors should consider using PheCode to define disease, such as

<https://phewascatalog.org/phecodes icd10>

We thank the reviewer for this suggestion. We now report single-protein regression results on PheCodes that intersect with previously selected ICD10 codes in Supplementary Table 12. In short, replacing individual ICD10 codes with PheCodes produces similar results, and the proteins found associated are the same. We have added the following paragraph to the methods:

"A disease may be manifested as several ICD codes, and broadening phenotype definition beyond single ICD codes could increase case numbers and thereby boost power. We use the PheCode Map 1.2 with ICD-10 codes to first select any PheCode whose definition included at least one ICD-10 code included in our previous analysis. We then used the union of cases for all ICD-10 codes included in those PheCode definitions to define case/control status, and repeated the analysis."

Regression outputs are presented in Supplementary Table 12 and confirm the single-protein results obtained with ICD-10 codes. A discussion of the additional protein signals discovered using the PheCode analysis is detailed in the Supplementary Text."

The corresponding Supplementary Text section is entitled "Additional Models produced by the PheCode analysis" and is not reproduced here for the sake of brevity.

8. The number of cases and controls should be reported for each disease in UKBB samples. There are a lot of related individuals in UKBB samples, the authors didn't mention how those were dealt with. Ignoring relatedness in samples could result in inflation.

We have now added a "number of cases" column in Supplementary Table 11b, similar to the one present in Supplementary Table 11a. We have also combined primary and secondary ICD10 codes in that table, so as to mirror our analysis, which considered the union of primary and secondary cases. This reduces the number of UK Biobank phenotypes to 80 (from 86).

In MR analyses, related individuals were excluded or corrected for by the analysts performing associations, however, we included related individuals in our predictive analyses in UK Biobank. In order to estimate the effect of relatedness on our results, we perform a sensitivity analysis, repeating the models excluding related individuals. As we now write in the Methods section on polygenic scoring, *"Finally, related individuals are present in UK Biobank, which might result in test statistic inflation when predicting disease risk. We perform a sensitivity analysis excluding related individuals, which produces similar results (Supplementary Text)".* Detailed statistics are given in Supplementary Text, section "Sensitivity Analysis".

9. The authors stated that they evaluate the predictive power of proteins for complex disease. But they performed only association analyses. AUC or R-squared should be used to evaluate prediction power in addition to p-value from association analyses.

We thank the reviewer for bringing this point to our attention. For protein prediction accuracy in Pomak, we have now included R-squared in Supplementary Table 10. In order to provide an accurate estimation, we perform three repeats of 5-fold cross-validation with a 20% hold-out set, and report both average and standard-deviation of R-squared on the training set, and a point R-square estimate on the hold-out set. For logistic regression models in complex disease risk prediction, R-squared cannot be computed like in a regular linear regression. We now report McFadden's pseudo-R-squared in addition to the AUC of the alternative and null models for our single-protein linear regressions.

We also report AUC and DeLong's test for AUC difference for the multi-protein models.

Reviewer #1 (Remarks to the Author):

I am comfortable with the revised manuscript.

Reviewer #2 (Remarks to the Author):

All my questions are addressed. I don't have further comments.

Reviewer #3 (Remarks to the Author):

The authors have adequately addressed my previous comments. I don't have any further questions.